# Causes of and Alternatives to Medication for Behaviours That Challenge in People with Intellectual Disabilities: Direct Care Providers’ Perspectives

**DOI:** 10.3390/ijerph19169988

**Published:** 2022-08-13

**Authors:** Shoumitro (Shoumi) Deb, Bharati Limbu, Gemma L. Unwin, Tim Weaver

**Affiliations:** 1Department of Brain Sciences, Faculty of Medicine, Imperial College London, 2nd Floor Commonwealth Building, Du Cane Road, London W12 0NN, UK; 2School of Psychology, University of Birmingham, 52 Pritchatts Road, Room 314, Edgbaston, Birmingham B15 2TT, UK; 3Department of Health & Social Care, School of Health Social Care and Education, Middlesex University, London NW4 4BT, UK

**Keywords:** people with intellectual disabilities, the causes of behaviours that challenge, alternatives to medication for behaviours that challenge, social care services, support staff, service/home managers, trainers

## Abstract

Behaviours that challenge (BtC), such as aggression and self-injury, are manifested by many people with intellectual disabilities (ID). National and international guidelines recommend non-pharmacological psychosocial intervention before considering medication to address BtC. Support staff play a pivotal role in the prescription process. Using coproduction, we developed a training programme for support staff, called SPECTROM, to give them knowledge and empower them to question inappropriate prescriptions and ask for the discontinuation of medication if appropriate and instead look for ways to help people with ID when they are distressed without relying on medication. We have presented data from two focus groups that we conducted during the development of SPECTROM: one that included support staff, and another that had service managers and trainers. In these focus groups, we explored participants’ views on the use of medication to address BtC with a particular emphasis on the causes of and alternatives to medication for BtC. Along with the participants’ views, we have also presented how we have addressed these issues in the SPECTROM resources.

## 1. Introduction

Behaviours that challenge (BtC), or challenging behaviour, can be defined as “culturally abnormal behaviour of such an intensity, frequency or duration that the physical safety of the person or others is likely to be placed in serious jeopardy, or behaviour which is likely to seriously limit the use of, or result in the person being denied access to, ordinary community facilities” [1]. BtC is common in people with intellectual (learning) disabilities (ID), with up to 60.4% of adults with ID showing at least one form of BtC [2,3,4]. BtC includes aggression, destructive behaviour, and self-injurious behaviour [5]. BtC can be difficult to manage and may lead to exclusion from community facilities, community placement breakdown and hospitalisation, and the use of restrictive practices such as physical restraint and inappropriate medication use [2,5].

To help a person with BtC, it is crucial to understand the reason behind the BtC rather than to inappropriately use medication [2,6]. BtC can be considered a form of communication whereby a person with ID conveys their distress [7]. For example, if someone is in pain or frustrated because of the demand put on them, they may shout and scream if they cannot communicate their distress to others. According to Matson and colleagues [8], the function of BtC could be categorised under six headings: attention (receive attention), escape (avoid something), non-social (factors internal to the person), physical (physical problems such as relief from pain), and tangible (achieve something). The function of the behaviour is assessed using functional behavioural analysis through Antecedent (situation before the BtC), Behaviour (the description of the actual behaviour), and Consequences (the consequences of the behaviour) ABC charts [9].

Both pharmacological and non-pharmacological psychosocial interventions such as positive behaviour support (PBS) [10] have been used to manage BtC [11,12]. A recent meta-analysis found a significantly long-lasting moderate overall effect of non-pharmacological interventions on BtC (effect size = 0.573) [13]. Interventions combining mindfulness and behavioural techniques, such as a PBS approach, showed greater impact than other interventions. Other PBS-based non-pharmacological interventions have also been shown to be effective in reducing BtC when compared with treatment as usual [14]. Another recent study has shown that a significantly higher number of adults with intellectual disabilities managed to come off their psychotropic medications when PBS was implemented in comparison to the group where no PBS was implemented [15].

Despite the poor evidence for the effectiveness of medications in managing BtC [6], psychotropic medications are used widely among people with ID (49–63%) and often off-license [16], which is a major public health concern [17].

Support (care) staff play a pivotal role in influencing the prescription process, such as by asking doctors to prescribe medication for BtC to begin with, and, given the lack of evidence of the effectiveness of the medications, are overly optimistic about the medication’s potential effect [18,19]. Support staff also are most anxious and obstructive to psychiatrists’ attempts to withdraw antipsychotic medication when appropriate [20]. Previous surveys of support staff in Australia [21] and the Netherlands [22] showed that most staff felt the use of psychotropic medications for BtC is justified. In our recent focus groups of support staff in the UK, some staff felt the use of medication is appropriate, whereas others felt that it is a ‘chemical restraint’ [23]. Among other factors, poor staff training and organisational policies are crucial factors in successfully withdrawing psychotropic medications [19,20,21]. Proper training and support for support staff are thus particularly important for successful programmes concerning the rationalisation of psychotropic medication use in adults with ID.

Training programmes are shown to be useful in many psychiatric disorders, including schizophrenia (30 RCTs) [24] and bipolar disorder [25], in improving patients and their caregivers’ quality of life (QoL). They have also been found useful in different neurodevelopmental disorders. For example, our recent meta-analysis has shown a moderate effect size of parental training on improving autism symptoms in children with autism spectrum disorder (ASD) [26]. A review of training programmes directed at parents and teachers included four RCTs and found improvement in attention deficit hyperactivity disorder (ADHD) symptoms with effect sizes of 0.05 to 0.77 [27]. A Cochrane review showed that training support staff helped to reduce antipsychotic prescriptions in people with dementia by 40–50% [28]. This was reflected in a nationwide trend in the reduction of antipsychotic prescriptions by 11% over 10 years (2005–2015) in the UK [29].

We have addressed the aforementioned issues by developing online training resources implemented through face-to-face interactive workshops for support staff caring for adults with ID in community settings. The training programme, SPECTROM (Short-term Psycho-Education for Carers To Reduce Over Medication of people with intellectual disabilities) (https://spectrom.wixsite.com/project accessed on 12 July 2022), was developed using a co-production method [30,31]. This was achieved by putting stakeholders’ experiences at the centre of the study and ensuring close and equal collaboration among them from the outset. The ultimate aim of SPECTROM is to empower, inform, and equip support staff with skills to understand the person they support, manage their own psychological responses to their behaviour, negotiate the care pathway, advocate on behalf of the person they care for, and take the views of adults with ID fully into account. The intended goal is to reduce requests from staff for medication and encourage staff to ask the prescribers questions about the necessity of the continued use of psychotropic medication, which should lead to a significant reduction in the use of medication and an increase in psychosocial interventions instead.

As part of the development of SPECTROM training, focus groups were conducted to explore staff perceptions on the use of psychotropic medication to manage BtC in people with ID and gather suggestions regarding the contents and format of SPECTROM. The aim of this paper is to present the findings of the first set of focus groups where support staff, service/home managers, and PBS trainers discussed their perceptions of and views on the use of psychotropic medications to address BtC in adults with ID. In this paper, we have presented two themes related to BtC: participants’ views on ‘the causes of BtC’ and ‘the alternatives to medication for BtC’.

## 2. Materials and Methods

We conducted two focus groups: one involving support staff and another with house/service managers and PBS trainers. These focus groups explored participants’ views on the use of psychotropic medication for BtC in adults with ID with a particular emphasis on the potential ‘causes of BtC’, and ‘alternatives to medication for BtC’.

### 2.1. Participants

We invited nine support staff, seven of whom agreed to attend, but eight ultimately took part. Five service/house managers and three trainers were invited, and all took part in the focus groups. We have not collected any demographic data on the participants.

### 2.2. Conduct of the Focus Groups

A topic guide (see Appendix A) was developed based on the literature review findings and the project’s aims and objectives. After discussion with the core team (BL, SD, TW, and GU) and other relevant stakeholders, the topic guide was finalised. This was employed flexibly and was open to emergent themes but framed using the Theory of Planned Behaviour model [32], thus examining beliefs and attitudes (e.g., about psychotropic medication and alternative approaches such as PBS) and how these might influence behaviour (e.g., in terms of requesting support from professionals to prescribe medication or provide help with alternative approaches). The interviews started with an exploration of participants’ experiences dealing with BtC in adults with ID and the use of medication for BtC. Then, the topic guide moved into issues related to potential causes of BtC, including physical, psychiatric, and environmental causes. From there, the topic guide entered into the issue of participants’ views on alternatives to medication, particularly psychosocial interventions such as PBS, for BtC. The sample size was a pragmatic decision. No formal sample size calculation was required for this study. This was a small study with limited resources and included primarily qualitative data collection. The minimum sample size we aimed for was 6–8 participants in each focus group. Participants were purposively sampled for each group to include a range of support staff in terms of their experience and from different organisations.

A researcher (BL) with previous experience in conducting qualitative research ran the focus groups with the help of the chief investigator (SD) under the supervision of an expert in qualitative research (TW). We used the approach utilised in our previous studies of interviewing the carers of people with ID as well as head injury [33,34,35]. Any paid carers, service/home managers, and trainers working with people with ID who showed BtC were eligible to participate in the study. A research advertisement was sent through the UK Voluntary Organisations Disability Group (VODG), an umbrella organisation of more than 35 social care service providers in the UK (social service, voluntary, and independent sectors). Nine service provider organisations agreed to take part, but, ultimately, only eight got involved. The organisations are Mencap, Challenging Behaviour Foundation, Achieve together, AT-Autism, Avenues Group, Dimensions UK, Milestones Trust, and National Autistic Society. Each organisation identified one available manager and trainer for the focus groups, and each manager identified two support staff. The focus groups were held face-to-face at a venue in London, England in March 2019. Eligible participants were then sent a SPECTROM study summary and information on the study. Once participants agreed, written informed consent was taken. Two separate focus groups were conducted, one with support staff and the other with service managers and trainers. Each lasted for approximately 90 min. The same topic guide was used for both groups, and a semi-structured interview was carried out. The focus groups were recorded using pseudonyms and professionally transcribed. Two authors (BL and SD) independently analysed the data to achieve a consensus. In the focus groups, care staff, service managers, and trainers explored issues around their perception and views on medication use for and causes and assessment of BtC.

### 2.3. Thematic Analysis

The transcriptions were analysed using thematic analysis. Thematic analysis is the process of identifying patterns and themes within qualitative data and texts. The data are interpreted to examine underlying meaning and ideas. Thematic analysis can be top-down deductive, driven by specific research questions, or bottom-up inductive, driven by data [36]. We used a top-down deductive approach to analyse the gathered data based on the topic guide and research questions we developed. NVivo 12 plus for windows software [37] was used to manage and analyse the data. The authors first familiarised themselves with the data. The transcripts were then read and interpreted for meaning and significance. Then, a code was given to each interpretation, and an initial coding frame was developed to organise the identified codes. This initial frame included main themes under ‘physical cause of BtC’, ‘psychiatric cause of BtC’, ‘environmental cause for BtC’, and ‘psychosocial interventions for BtC’. The coding frame developed was continuously reviewed and refined as new codes emerged or as it was searched for patterns. The data and quotes were indexed on identified codes or new emerging codes. The coding framework was then searched for patterns for emerging themes. Similar codes were categorised together to produce patterns and themes. The identified categories were also reviewed to ensure the emerging category was discrete and modified as necessary. The identified themes were then reviewed and revised if needed. Once no new themes emerged, the themes were finalised and defined. The themes and analysis process were also overseen and verified by two authors (GU and TW) who were experienced qualitative researchers.

## 3. Results

We have presented two themes and six subthemes from the thematic analysis of focus group discussions: (a) causes of BtC, including psychiatric disorders, and (b) alternatives to medication including PBS, person centred approaches, understanding the person, developing relationships, and collaborative working. In the results section, we present the main themes and subthemes followed by relevant quotes directly taken from the transcripts. We have used codes in the parenthesis such as ‘SS’ for support staff, ‘SM’ for service/house manager, and ‘TR’ for trainer to identify the quotes from the different groups of participants.

## 4. Causes of BtC and How Understanding Them Can Help the Person with ID

Participants discussed the causes of BtC and how understanding them would help address BtC using alternatives to medication. The central emerging theme was ‘behaviour is a means of communication’. Triggers for BtC were discussed, including the environmental factors, particularly if the people are not placed in the right environment, and the help they need to accommodate any changes in their life. Participants felt it would be good to have a checklist to recognise the factors that lead to BtC. We have addressed this issue by developing a Checklist for the Assessment of Triggers for the behaviour of concern Scale (CATS) [38]. Staff can use this list to identify the triggers for BtC, which will help complete the Antecedent-Behaviour-Consequence (ABC) chart and functional behaviour analysis.

“*… when she was aggressive, and all she wanted was the curtains tied back. So, what someone else would think of that, you know it’s just, it’s just knowing.*”(TR)

“*I think it was literally a checklist of going through the whole thing with her. What could it be?*”(SM)

“*To be considered, the environment, how they feel in the environment, do they feel comfortable there?*”(SS)

“*… placement staff comes into it a lot on whether someone who is in the right placement.*”(SM)

Some participants felt it is not helpful when people come with a label such as being very aggressive. They thought they should thoroughly reassess the person and their behaviour to draw a new person-centred support plan.

“*Looking at the support plan. I mean had to rewrite her support plan with in the space of two weeks because she came labelled as being a very difficult person, very challenging. And naturally she was scared so her way of reacting was to fight. And the treatment assessment unit, she was in where I visited her, I was scared, so she must have been absolutely petrified.*”(SM)

Participants discussed how physical problems, such as pain in the body, can manifest as BtC and be wrongly treated with antipsychotics.

“*I know for a fact people have been given anti-psychotic medication and actually it’s something physical that’s wrong. They’ve got toothache or they’ve got tummy ache or they don’t like the colour of their room. Something as simple as that, you know.*”(SM)

“*Um, but recently I had someone who was exhibiting behaviour but it was because he had chest infection and he was in pain but he couldn’t vocalise or tell someone I’m in pain. So, then staff log it as challenging behaviour, we need to talk to a psychiatrist.*” (SM)

The issue of communication came up in the discussion. SPECTROM has a module on effective communication and another on effective engagement with people with ID. Both modules encourage staff to learn the best ways to communicate with the person they support so that they can concentrate on their skill-building rather than BtC.

“*I think often people are perceived as showing behaviours because of the condition, but actually its external factors and things in the environment or how they’re being supported. Um, the communication strategies that they’ve got.*” (SM)

“*Sometimes when you have challenging behaviours it’s not, it’s because someone wants something. It’s like you said. Yeah, they can’t communicate. They’re telling you they want something done or to do something. Communication. It’s always communication.*”(TR)

### Causes of BtC: Psychiatric Disorder

While discussing the causes of BtC, the role of psychiatric disorders was raised. All participants unanimously agreed that no meaningful training currently exists on the issue of psychiatric disorders in ID, although there is an increasing awareness of Autism Spectrum Disorder (ASD).

“*… we’ve not really, we’ve never really given any sort of any specific training on psychosis or um, mine was just basically what I’ve read from you know carer plans and things like that. Um, but nothing, we’ve never ever attended any training on it.*”(SS)

Staff found it difficult to distinguish between psychiatric disorders and BtC. They felt that when the person is trying to communicate their needs through BtC, it is often misinterpreted as a psychiatric illness.

“*When is it challenging behaviour and when is it psychosis, like you say? When is it that he’s hearing voices that’s telling him to do something and when is it when he’s doing it because of his own accord that he’s trying to communicate something or maybe an expression?*”(SS)

“*… how much of what they’re displaying is because of their mental illness or how much is that just because they’re trying to tell you something and you’re not able to understand it? As support staff, something I struggle with personally is knowing the difference.*”(SS)

Participants felt that when there is a crisis and the person with ID displays BtC, it is often considered part of a psychiatric disorder. As a result, they do not explore other factors, including the environment, to check what is causing the behaviour.

“*… when he attacked staff but he didn’t hear voices, was that because of the psychosis or what that because he was having a, he was going through, he was going through that crisis.*”(SS)

There was significant confusion about the overlap between what may be ASD symptoms (e.g., living within an inner world) and psychiatric symptoms. SPECTROM has comprehensive modules on ASD and ADHD. The ASD module discusses the potential overlap between the ASD phenotype and psychiatric symptoms in detail, but acknowledges that many psychiatric disorders, including psychosis, are common in people with ASD [39].

“*… there was too much stimulation going on and that frustrated him so that he just lashed out. Because he’s also diagnosed with autism, there’s also that to consider as well. So that was due to the environment why he acted that way rather than it being psychosis because something told him to lash out.*”(SS)

Participants discussed the difficulty of distinguishing between trauma-related symptoms and psychiatric symptoms. Some participants thought that psychotic symptoms might, in fact, be the memories of past traumatic events spoken aloud rather than genuine psychosis. There is also the possibility of past traumatic experiences inducing real psychotic symptoms [40,41]. This issue has been discussed in the psychiatric disorder module of SPECTROM.

“*He believes that he hears people, and he will say that they made me do it and things. ….is that because he’s saying that but actually is that just a past traumatic event that’s happened that then he’s remembering it?*” (SS)

One support staff expressed frustration on the lack of awareness among some general practitioners who take a medical approach to BtC and say that these are often due to a psychiatric disorder.

“*… one of the GP I discussed it. But what do you expect? He’s got mental health problems. I’m talking about uh, educating people.*”(SS)

## 5. Alternatives to Medication

### 5.1. Positive Behaviour Support (PBS)

After discussing the possible causes of BtC, the discussion progressed to how to best address BtC without relying on medication. This started with a discussion around PBS and how this framework can be used to reduce inappropriate uses of medication. Some larger service provider organisations seem to have their own PBS support team, which staff found very helpful. However, even in big organisations, the PBS support team resource is not always adequate, and most smaller organisations will not have this kind of support available at all. Most staff in large organisations also only have basic training on PBS.

“*She got into crisis, was in crisis for months and regardless of whatever we were doing as a carer, as a care staff, and you know we brought in um, um positive behaviour support team. We were working really closely with them. We’ve got really, really robust PBS plans.*”(SS)

“*Part of our PBS was to give him space, so we gave him space.*”(SS)

“*So now that he’s come…..to our company, we said we were going to support him for positive behaviour support plan. …..we’ve seen a different person. He’s living a better life and that’s because of the service that he’s in that’s allowing him to live a better life.*”(SS)

“*You know, staff have PBS training, um, we work with behavioural support analysts that come in and collect data.*”(SM)

The issue of the cost of providing the service was discussed. One trainer highlighted the fact that using the PBS approach can reduce the frequency of BtC and the use of medication, which will save money in the long run. Therefore, a case could be made for social and health care commissioners to invest the funds needed to implement PBS now in order to save money in the future.

“*So that’s got big budget implications as well, because certainly similar examples where somebody has come out of long stay and they were on, you know, X amount of medication and all the support plans are saying three to one support in our community. ….but two years down the line that three to one support becomes one to one support because somebody’s taken a holistic approach to look at what do we need to do to support this person?*”(TR)

### 5.2. Person-Centred Care Approach

Other alternatives to medication based on the PBS approach were discussed, including a person-centred care approach. There seems to be more awareness within the larger social care provider organisations in the UK and their staff about using alternatives to medication to address BtC through a person-centred care approach.

“*So our staff are very aware of the, um, other ways to help behaviours rather than prescribing medication.*”(SM)

“*Right now, there has been a lot of awareness that’s been created that people are using all those alternatives rather than medication.*”(SM)

“*… He’s more engaged. He’s more willing to communicate, willing to do things. Willing to participate because we are allowing him to do.*”(SS)

### 5.3. Understanding the Person

Another person-centred care approach the participants discussed was understanding the person behind the behaviour. Participants felt that, once they got to know the person well, it became easier to help them without using medication when they were distressed. Sometimes people come from a previous placement with a label of being aggressive, but staff felt that this should not deter them from trying to get to know the person and help them with their skill development instead of concentrating on their BtC. One service manager mentioned that, if you do not understand the person and the reason for their BtC, you tend to blame the person for their behaviour which does not help.

“*Um she’s not written up for anything, PRN at all so her behaviour are managed through um, staff really knowing her well.*”(SS)

“*So, if you don’t get to know what they are on about, then they will start to display challenging behaviour.*”(SS)

“*And you find out what they like. Nobody asked them. And show positive interest in what they like and you know to talk to them like they’re a real human being.*”(SS)

“*Instead of taking the baggage of their history with them. And it’s about stripping all of that out, and almost starting again and saying let’s look at this person as a whole now and get the right people involved.*”(TR)

“*And I believe that does have a lot to do with staff attitudes and behaviours. There are still some places where they blame the person and they are behaving because that’s the way they are, rather than then behaving in this way because someone has taken something from that person and they don’t know how to control it.*”(SM)

### 5.4. Developing Relationships

Another vital person-centred care approach that involved developing a positive relationship with the person they support was discussed. One participant said that, even after being scratched by the person with ID on one occasion, she continued to build a relationship with the person, which helped to improve the BtC.

“*So, I stayed longer and now we’ve got a bond that when I’m on annual leave it’s a problem because we build that relationship and it got to a point that everything, she wants to do has to be with me.*”(SS)

“*… where he became heightened and within that he was able to scratch me which left a scar. From that incident, I didn’t change my approach towards him but obviously I was cautious because I’m not trying to get injured. But through that, because I hadn’t changed the way I was, and no more the next day and even after that, after that incident he saw me at breakfast, I’ll still engage with him, still carried on as normal. The relationship that we have now is that you know he, he trusts.*”(SS)

“*But now we got to a point where I can take her to any appointment, we can sit down. So, I, we build a bond.*”(SS)

### 5.5. Collaborative Work

Participants felt that multi-disciplinary work was in the best interests of the person with ID. This includes other relevant professionals like doctors and nurses involved in the care of the person and the person with ID themselves and their families.

“*With the prescribing, it’s also working with families and psychiatrists, because sometimes there’s a parent who is going to tell you I know my child and I think they need this, they have been using this.*” (SM)

“*If the families are involved or any other person involved in their circle of support, they will also attend the meeting. And it’s a best interest meeting as well.*”(SM)

“*… um, how you work as a team with those different professionals. We have a pharmacy check.*”(SM)

One service manager mentioned that they asked key support staff to accompany the person with ID to the clinic for the medication review. She also emphasised the importance of gathering all relevant information before attending the clinic and involving the person with ID and their family, if possible. SPECTROM provides a checklist for staff to go through in preparation for a formal medication review and a set of questions that staff could or should ask the prescribers in the clinic.

“*… before we go to that meeting, let’s have a discussion first so there’s no surprises. What are we going to say, what are we presenting to the clinical psychiatrist that doesn’t know this person as well as what we do. What does the family members got to say about it? So, we go into that meeting with that individual, if they’re open to input and they have the capacity to do that, to feed into that meeting so that it’s productive for that person.*”(SM)

## 6. Discussion

In this paper, we have presented a wealth of data on wide-ranging issues from focus groups, primarily on participants’ perception of the causes of BtC and methods to help people with ID without using medication (alternative to medication).

### 6.1. Causes of BtC Including Physical and Psychiatric Disorders

As for the causes of BtC, the discussion primarily revolved around physical disorders, psychiatric disorders, and environmental factors. There was a consensus among the participants that BtC is often a means of communication and that the causes and effects of BtC need a thorough assessment to draw up an effective person-centred behaviour support plan to reduce overreliance on medication to address BtC. In SPECTROM, we have provided a module on the assessment of behaviour and examples of two assessment schemas: **B.M.P.P.S.** (assessment of the **B**ehaviour itself, **M**edical issues, the **P**erson with ID, **P**sychiatric/psychological issues, and **S**ocial issues) [5] and **H.E.L.P.** (assessment of **H**ealth and medical conditions, **E**nvironment and support, **L**ived experience and emotional well-being, and **P**sychiatric illness) [42].

Physical disorders are more prevalent in people with ID than in the general population [43]. Although it is not uncommon for physical problems that are common in people with ID, such as pain in the body (headache, toothache), constipation, and acid reflux, to lead to BtC, particularly in people who cannot communicate their feelings, it is not always easy to identify these causes. Therefore, a high degree of suspicion and thinking about a physical cause for BtC should help ensure the correct diagnosis. If necessary, a therapeutic trial with a painkiller where pain is suspected may help. We have developed a ‘Physical disorder’ module in SPECTROM and provided examples of proformas that could be used to detect and rate the severity of pain. Another problem mentioned was the difficulty of carrying out required investigations such as blood tests, X-rays, etc., for many people with ID. SPECTROM provides practical guidelines for addressing these issues, such as using accessible information including pictures of the tests, teaching the person relaxation techniques, etc. SPECTROM also provides videos on ‘what happens during a health check’, ‘what happens during a blood test’, ‘what happens during an MRI’, etc., to familiarise people with ID before they go for any of these investigations.

The issue of the relationship between psychiatric disorders and BtC is a complex one [39]. Most staff did not receive any training or information on this. Participants raised the question of how to distinguish between psychosis and BtC due to environmental factors or when caused by traumatic life events. Therefore, SPECTROM has developed a psychiatric disorder module to provide information to support staff.

The participants discussed the issue of overlap between ASD and BtC. ASD and ID commonly co-occur; 38% of people with ASD have ID, and a similar proportion of people with ID are also known to have ASD. ADHD is equally common in both ASD and ID, although this issue was not discussed in the focus groups. The extensive overlap of symptoms between ID, ASD, and ADHD often makes it difficult to tease apart these diagnoses in people with ID [39,44,45]. As BtC is common in ID (18–60%) [2], ASD (10–15%) [46], and ADHD [44], respectively, the rate of BtC increases when all three conditions co-exist [39]. Therefore, SPECTROM has developed comprehensive modules on ASD and ADHD, with information on strategies to help people with ID and ASD and/or ADHD to reduce BtC. Currently, there is more awareness of the trauma-induced symptoms in people with ID [40], some of which may manifest as psychiatric disorders or BtC [40,41]. Many support staff in large provider organisations seem to be more conscious of this possibility when they draw up a person-centred care plan for the people they support.

### 6.2. Alternatives to Medicine through the Use of Person-Centred PBS Care Planning

After discussing the causes of BtC, participants progressed to discuss how to address BtC, particularly by using a psychosocial approach and avoiding medication use. The most well-known psychosocial approach is PBS [10], which provides a framework to apply in practice [47]. Most staff from large provider organisations seem familiar with PBS principles, which they use to draw person-centred care plans [48] for people they support. This approach seems to reduce the need to use medication to address BtC. One study found that community team-led implementation of PBS framework did not reduce BtC in community settings [49]. However, others found that PBS and other psychosocial approaches were useful in reducing aggression among people with ID [14].

Participants acknowledged that understanding the person they support is key to understanding BtC and improving them without relying on medication. Service managers and trainers mentioned not focusing on the ‘labels’ people with ID bring with them but concentrating on the whole person in order to understand their likes and dislikes, strengths and weaknesses, and develop a positive relationship. All participants agreed on a holistic approach to care planning and addressing BtC.

### 6.3. Shared Decision Making

All participants in the focus group agreed on the need for multidisciplinary work and acknowledged that there is scope to improve shared decision-making by involving the person with ID and their families. In a previous study, we found that adults with ID did not feel sufficiently involved in making the decision about their medication [50]. Some adults with ID who we interviewed were dissatisfied with their medication, mainly due to lack of involvement in the treatment decision, adverse effects, lack of efficacy, and a ‘desire to lead a normal life’. In another study, most of the six adults with ID who were interviewed complained about not having enough information on their medication, particularly in an accessible format [51].

A similar sentiment has been echoed by family members, many of whom felt that there is an over-reliance on medication instead of taking a holistic approach to address BtC [52]. In our study, many of the 20 family caregivers interviewed complained about not having enough information about the care of their loved ones and not having enough involvement in the decision-making process [53]. Family caregivers play an essential part in the care of people with ID, even when they do not live in the family home; they are the only constant presence in the life of the person with ID, whereas professionals and care staff come and go. Therefore, the knowledge of their loved ones is of paramount importance in care provisions for people with ID. To address these issues, we have devoted an entire module to ‘effective liaison with families’ in SPECTROM. This module teaches staff to respect family caregivers’ views, acknowledge family caregivers’ expertise, communicate with them without jargon, and include family caregivers and the person with ID in care planning, including prescriptions of medication, from the outset in order to promote more shared decision-making.

Although several studies explored care staff views on medication use, they rarely reported on staff perceptions of the causes of BtC, which was at the heart of the discussions within our focus groups. In our previous study, few care staff explicitly reflected that their own behaviour might influence aggressive behaviour in adults with ID [54]. Furthermore, only 16% of staff interviewed in this study mentioned issues around communication despite much of the aggressive behaviour being considered to be communicative [7]. Staff felt that they would benefit from training and information about potential triggers to help them think more about environmental conditions and their own role in precipitating BtC. Many other researchers have highlighted support staff frustration for not having the right training and their desire to gather more knowledge and training on (a) mental health issues, (b) medication prescribing (when to use them and why, and when not to use them and why), (c) medication side effects, (d) and when and how medication could be safely withdrawn [55]. All of these issues are addressed in the SPECTROM programme.

### 6.4. Strengths of the Study

Previous studies of staff surveys primarily concentrated on staff knowledge of psychotropic medication. They rarely explored their views on the causes of BtC, which our study has addressed. One strength of this study is that it examined not only support staff’s opinions, but also service managers’ and trainers’ opinions. Previous studies have primarily focused on the experiences and perceptions of support staff and family caregivers. Another strength of this study is that support staff, service managers, and trainers were interviewed separately to avoid the influence of service managers on support staff responses. The support staff, managers, and trainers were also from different service provider organisations, allowing us to capture different aspects of the participants’ experiences. In addition, the anonymised data analysis allowed support staff to express their opinion freely, thus increasing the face validity of our findings. Another strength was that the data were analysed by two authors.

### 6.5. Limitations of the Study

All participants were recruited from large service provider organisations. Therefore, the views and experiences of staff working in smaller organisations were not captured, which may have differed significantly. Another weakness is that service managers identified the support staff for the focus groups, thus managers may have chosen support staff that were particularly eager to get their voices heard on this topic.

## 7. Conclusions

Our exploration of the views of support staff, service managers, and trainers on the causes and management of behaviours that challenge in people with intellectual disabilities used focus groups to reveal that most participants have some knowledge of the physical, psychological, and environmental causes of BtC. Most were also familiar with the concept of positive behaviour support that could be used to support people with intellectual disabilities who display behaviours that challenge without relying on medication. However, it is worth remembering that all participants in our study were employed by large social care service provider organisations in the UK. Most of these organisations are likely to have their own positive behaviour support teams; smaller provider organisations are unlikely to have this in-house support. Participants unanimously expressed concern about their lack of knowledge of psychiatric disorders and their relationship with behaviours that challenge in people with intellectual disabilities. There were no major discrepancies in the views of the support staff and those of the service managers and PBS trainers. All parties have agreed on the need for more information on medication, their indications, and side effects. We have addressed these issues and others raised by the participants in SPECTROM modules.

## Data Availability

Transcripts of the focus group discussions are available from the authors subject to approval from the study sponsors and the funders.

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
