# Peer review of "Causes of and Alternatives to Medication for Behaviours That Challenge in People with Intellectual Disabilities: Direct Care Providers’ Perspectives"

_ijerph, 2022, doi:10.3390/ijerph19169988_

Round 1

Reviewer 1 Report

Thank you for the opportunity to review the manuscript "Support staff, service managers and trainers' perception of the causes of behaviours that challenge in people with intellectual disabilities and alternatives to medication to address them." This study focuses on finding alternative treatments to problem behaviors - an important topic in the field of intellectual disability. I believe this paper will make a good contribution to the field. However, some changes must be made before the study is suitable for publication. Please find my suggestions below.

Title: consider shortening

Introduction

Avoid using "etc." on page 1, line 34

Page 1, first sentence of second paragraph - provide reference

Page 1, second paragraph - provide examples of functions of  behavior (i.e., sensory, escape, attention, tangible)

General: A mention/description of Applied Behavior Analysis (ABA) is warranted

Page 2 - additional details/more specific information needed when describing (1) positive effects of PBS on addressing BtC and (2) support staff role in the decision to consider medication.

Page 2, line 67 - rephrase "prescribing process."  Be specific in their role in the decision to medicate.

Page 2, line 84 - rephrase "This should" to "The intended goal...."

Methods

Page 3, line 130 - Separate into new paragraph beginning with "Two separate focus groups..."

Move the description of the participants to the Methods section. Additionally, a more robust description of participants is needed (how long have they worked in their role, etc.). When describing gender, rather than reporting the number of females, be specific. Did the remaining participants identify as male? As more people identify as non-binary, it is best to be specific (and inclusive, if applicable) in reporting.

Results

The themes appear well-founded, given the quotes provided. However, in some cases there is an over abundance of quotes. Be selective to find quotes that are truly representative of the theme being described.

The concept of Antecedent-Behavior-Consequence (ABC) and functional behaviour analysis appears on page 4. Provide a description of these concepts in the intro - ideally with a description of ABA.

Remove any discussion of SPECTROM in the Results section. This should be described in the Discussion section, in relation to themes that emerged from focus groups.

This is not the fault of the authors, but the formatting throughout the Results section was off (text was right-aligned in my copy).

Discussion

Page 11, line 477 - change "wealthy data" to "a wealth of data"

Shorten 4.1 to "Causes of BtC"

Page 12, line 531 - remove "the" in "the PBS"

Page 13 - expand discussion on SPECTROM's focus on family collaboration.

Page 13 - High inter-rater reliability is mentioned as a strength. This should be noted in the Methods section, including specific percentage of agreement (if calculated).

Again, thank you for the opportunity to review this paper. With these changes in place, I believe this study would make a fine addition to IJERPH.

Author Response

Reviewer 1

Reviewer comment: Title: consider shortening

Our response: We have now shortened the title to: “Causes of and alternatives to medication for behaviours that challenge in people with intellectual disabilities: direct care providers’ perspective.”

Reviewer comment: Avoid using "etc." on page 1, line 34

Our response: We have now removed “etc.”

Reviewer comment: Page 1, first sentence of second paragraph - provide reference

Our response: We have now provided the references.

Reviewer comment: Page 1, second paragraph - provide examples of functions of behavior (i.e., sensory, escape, attention, tangible)

Our response: We have now added reference to Matson and colleagues’ (2012) six functions of behaviour as proposed in QABF, namely attention (receive attention), escape (avoid something), non-social (factors internal to the person), physical (physical problems such as relief from pain), and tangible (achieve something).

Reviewer comment: General: A mention/description of Applied Behavior Analysis (ABA) is warranted.

Our response: We have now briefly mentioned the ABC chart and functional behavioural analysis notwithstanding the limited space in the manuscript. 

Reviewer comment: Page 2 - additional details/more specific information needed when describing (1) positive effects of PBS on addressing BtC and (2) support staff role in the decision to consider medication. 

Our response: We have now added two paragraphs to cover these two areas supported by appropriate references notwithstanding the limited space in the manuscripts.

Reviewer comment: Page 2, line 67 - rephrase "prescribing process."  Be specific in their role in the decision to medicate.

Our response: We have now described in more detail with accompanying references the “prescribing process”, such as the initiation of prescriptions and discontinuation of them.

Reviewer comment: Page 2, line 84 - rephrase "This should" to "The intended goal...."

Our response: This sentence is rephrased as advised by the other reviewer.

Methods

Reviewer comment: Page 3, line 130 - Separate into new paragraph beginning with "Two separate focus groups..."

Our response: We have now revised this section to clarify the process of the focus groups.
Reviewer comment: Move the description of the participants to the Methods section. Additionally, a more robust description of participants is needed (how long have they worked in their role, etc.). When describing gender, rather than reporting the number of females, be specific. Did the remaining participants identify as male? As more people identify as non-binary, it is best to be specific (and inclusive, if applicable) in reporting.
Our response: We have now moved the description of the participants under the Methods section. We have now declared that we have not collected any demographic data on the participants and removed any references to their genders.
Results
Reviewer comment: The themes appear well-founded, given the quotes provided. However, in some cases there is an over abundance of quotes. Be selective to find quotes that are truly representative of the theme being described.
Our response: We have taken this comment into account and removed many quotes. However, as another reviewer appreciated reading the quotes as they helped with the flow of the Results section, we had to strike a balance on this. 
Reviewer comment: The concept of Antecedent-Behavior-Consequence (ABC) and functional behaviour analysis appears on page 4. Provide a description of these concepts in the intro - ideally with a description of ABA.

Our response: We have now mentioned functional analysis, ABC and ABA in the introduction section. 

Reviewer comment: Remove any discussion of SPECTROM in the Results section. This should be described in the Discussion section, in relation to themes that emerged from focus groups.

Our response: We have removed the discussion of SPECTROM from the results section.

Reviewer comment: This is not the fault of the authors, but the formatting throughout the Results section was off (text was right-aligned in my copy).

Our response: We are sure that the journal editorial office will take care of this!

Discussion

Reviewer comment: Page 11, line 477 - change "wealthy data" to "a wealth of data"

Our response: We have changed this.

Reviewer comment: Reviewer comment: Shorten 4.1 to "Causes of BtC"

Our response: We have significantly reduced this section. However, we have now included here a few additional SPECTROM discussion points from the Methods section. 

Reviewer comment: Page 12, line 531 - remove "the" in "the PBS"

Our response: We have now removed “the” before “PBS.” 

Reviewer comment: Page 13 - expand discussion on SPECTROM's focus on family collaboration.

Our response: We have expanded the discussion on family collaboration.

Reviewer comment: Page 13 - High inter-rater reliability is mentioned as a strength. This should be noted in the Methods section, including specific percentage of agreement (if calculated).

Our response: We have now moved this statement to the Methods section. As we don’t have exact data on inter-rater reliability, we have removed this statement from the ‘Strengths’ section.

Reviewer comment: Again, thank you for the opportunity to review this paper. With these changes in place, I believe this study would make a fine addition to IJERPH.

Our response: Thanks for the comments. We have found the comments very useful.

Reviewer 2 Report

The authors take up a very important issue of coping with difficult behaviors of people with intellectual disabilities. Identifying potential environmental resources in this regard is an essential step in planning support solutions. The article undoubtedly has cognitive value. In my opinion, the main problem of this work is the lack of a specific goal of the article itself, but also of the research whose results are presented by the authors. The authors describe the results of their own research, but also the assumptions and content of their program (SPECTROM), which I believe was based on, inter alia, the analysis of environmental resources, and thus also the results of the presented research conducted by the authors.

Detailed comments:

1.      I propose to modify the keywords, they should clearly and specifically reflect the subject matter of the work. They should also express its most important aspects: people with intellectual disabilities; the causes of challenging behaviors; alternatives to medication for challenging behaviors; social care services, support staff; service/home managers; trainers.

2.      Please complete the Introduction with references to E. Emerson, who has contributed significantly to understanding the issue in question.

3.      Please specify the research goal and problems.

4.      Please clarify the matter: on p. 3 the authors write about 4 focus groups. Two of them were to discuss the SPECTROM program. Results show that two groups share their opinions on difficult behaviors (p. 3 also contains information about two groups, “Two separate focus groups were conducted, one with support staff and one with service managers and trainers” p. 3).

5.      The authors focus on two topics that resulted from the thematic analysis. This approach is rare in qualitative research. All the themes that emerged during data analysis constitute a whole that describes the essence of the phenomenon. I believe that a lot of space in the work is devoted to the description of the program - also in the description of the results. It is impossible to present research results and the program simultaneously. I propose to present all the results and mention the program in the Conclusions, devoting a separate article to it. Interjected program descriptions create cognitive chaos.

6.      I propose to avoid categorical phrases such as, “Ours is the first study that examined support staff, service managers, and trainers’ perceptions of the causes of BtC and the use of psychotropic medication for BtC.” In the context of many literature resources and often difficulties reaching them all, such statements may be ungrounded.

7.      In the case of qualitative research, it is difficult to talk about generalization, even if the number of cases increases (see 13).

Author Response

Reviewer 2 

The authors take up a very important issue of coping with difficult behaviors of people with intellectual disabilities. Identifying potential environmental resources in this regard is an essential step in planning support solutions. The article undoubtedly has cognitive value. In my opinion, the main problem of this work is the lack of a specific goal of the article itself, but also of the research whose results are presented by the authors. The authors describe the results of their own research, but also the assumptions and content of their program (SPECTROM), which I believe was based on, inter alia, the analysis of environmental resources, and thus also the results of the presented research conducted by the authors.

Detailed comments:

Reviewer comment: 1.      I propose to modify the keywords, they should clearly and specifically reflect the subject matter of the work. They should also express its most important aspects: people with intellectual disabilities; the causes of challenging behaviors; alternatives to medication for challenging behaviors; social care services, support staff; service/home managers; trainers.

Our response: We have now amended the keywords in light of this comment.  

Reviewer comment: 2.      Please complete the Introduction with references to E. Emerson, who has contributed significantly to understanding the issue in question.

Our response: We have now provided Emerson’s reference to define Behaviours that challenge. 

Reviewer comment: 3.      Please specify the research goal and problems.

Our response: We have now added the aim of the study at the end of the Introduction section.

Reviewer comment: 4.      Please clarify the matter: on p. 3 the authors write about 4 focus groups. Two of them were to discuss the SPECTROM program. Results show that two groups share their opinions on difficult behaviors (p. 3 also contains information about two groups, “Two separate focus groups were conducted, one with support staff and one with service managers and trainers” p. 3).

Our response: We have now clarified this at the beginning of the Methods section.

Reviewer comment: 5.      The authors focus on two topics that resulted from the thematic analysis. This approach is rare in qualitative research. All the themes that emerged during data analysis constitute a whole that describes the essence of the phenomenon. I believe that a lot of space in the work is devoted to the description of the program - also in the description of the results. It is impossible to present research results and the program simultaneously. I propose to present all the results and mention the program in the Conclusions, devoting a separate article to it. Interjected program descriptions create cognitive chaos.

Our response: We agree with the reviewer. However, the emerging themes although interconnected but could be categorised under two distinct subthemes, one including ‘the causes of behaviour’ and ‘alternatives to medication to manage behaviour’, and the other one including ‘the current trend in medication use’ and ‘withdrawal of medication.’ The first subtheme covers the “behaviours,” and the second subtheme covers the “medication.” Each of them needs a separate Introduction and Discussion. Each of them has a wealth of information including many subthemes which would be difficult to fit within one paper given the restricted space. Also given the distinct nature of these two themes presenting them together would have confused the readers and as a result, very important messages concerning these two distinct themes would have been lost.

Our response: We have now moved SPECTROM discussions from the Results section to the Discussion section.

Reviewer comment: 6.      I propose to avoid categorical phrases such as, “Ours is the first study that examined support staff, service managers, and trainers’ perceptions of the causes of BtC and the use of psychotropic medication for BtC.” In the context of many literature resources and often difficulties reaching them all, such statements may be ungrounded.

Our response: We have now removed this sentence. 

Reviewer comment: 7.      In the case of qualitative research, it is difficult to talk about generalization, even if the number of cases increases (see 13).

Our response: We agree that in qualitative research, we do not look for generalisability, rather we seek applicability of the results. In this study, this was confirmed in the co-production process. We have removed the sentence about the small number of participants is considered a limitation for generalisation.

Reviewer 3 Report

The paper discusses a relevant topic for both practice and science. I can imagine that a training - such as it has been developed – has enormous added value. However, I am constantly wondering what the current study is actually about. Also,the authors seem to struggle combining the incredibly rich qualitative data they have collected with the format of a scientific article with word limits.

In the introduction, the relevance of the study and the context are very clearly stated. However, I feel that the problem statement, research question and purpose of this specific paper are missing.

The title of the study aims at the perception staff has of the causes of BtC , however I think the intervention is also aimed at changing staffs attitude. I would be interested in seeing that emphasizes in the introduction. Also more evidence about ways to change perception would be interesting, eg. how psychoeducation works and could affect the behaviour of staff..

The introduction ends with the remark that we going to look into perceptions and views of care professionals. If that is so, the authors need to write towards this more in the introduction.

Reading the method, the selection of the respondents is clear. The analysis could be described more in detail, which codes were used (the authors describe a top down coding, did they start with a coding list?). What do the authors mean by patterns?  Did an inter or intra-rater reliability procedure take place?

The method lacks the topic list or questions that were asked in the focus group.

I would prefer to find a description of the participants in the method section.

in the results, the authors indicate that they only focus on the causes of BtC including psychiatric disorders. This choice should have flowed logically from the introduction and method. The results read beautiful and I like reading the quotes from the focus groups. As a result, the introduction and discussion will fit better together, they now seem to be separate from each other.

In the discussion and conclusion the authors should investigate the difference between support staff’s opinions and the opinion of service managers’ and trainers’.

Author Response

Reviewer 3 

The paper discusses a relevant topic for both practice and science. I can imagine that a training - such as it has been developed – has enormous added value. However, I am constantly wondering what the current study is actually about. Also,the authors seem to struggle combining the incredibly rich qualitative data they have collected with the format of a scientific article with word limits.

Reviewer comment: In the introduction, the relevance of the study and the context are very clearly stated. However, I feel that the problem statement, research question and purpose of this specific paper are missing.

Our response: We have now added at the end of the Introduction section the specific aim of this study. 

Reviewer comment: The title of the study aims at the perception staff has of the causes of BtC , however I think the intervention is also aimed at changing staffs attitude. I would be interested in seeing that emphasizes in the introduction. Also more evidence about ways to change perception would be interesting, eg. how psychoeducation works and could affect the behaviour of staff.

Our response: We have now added discussions with references to the studies on staff training changing clinical practice as well as staff views on prescribing.

Reviewer comment: The introduction ends with the remark that we going to look into perceptions and views of care professionals. If that is so, the authors need to write towards this more in the introduction.

Our response: We have added further information with appropriate references on the staff’s views on psychotropic prescribing. 

Reviewer comment: Reading the method, the selection of the respondents is clear. The analysis could be described more in detail, which codes were used (the authors describe a top down coding, did they start with a coding list?). What do the authors mean by patterns?  Did an inter or intra-rater reliability procedure take place? 

Our response: We have added the main coding frames. We have not done an inter-rater reliability, so removed any reference to this from the text.

Reviewer comment: The method lacks the topic list or questions that were asked in the focus group.

Our response: We have supplied the topic list as an appendix in the supplement because of the lack of space in the text. However, we have mentioned some broad themes under the initial codes in the Methods section.

Reviewer comment: I would prefer to find a description of the participants in the method section.

Our response: The description has now been moved to the Methods section. 

Reviewer comment: in the results, the authors indicate that they only focus on the causes of BtC including psychiatric disorders. This choice should have flowed logically from the introduction and method. The results read beautiful and I like reading the quotes from the focus groups. As a result, the introduction and discussion will fit better together, they now seem to be separate from each other.

Our response: We have now added new paragraphs in the Introduction and the Discussion sections, primarily on the previous studies of staff views on the use of psychotropics for behaviours that challenge and also how staff training may help improve the rationalisation of psychotropic use, making the Introduction, Results and Discussion sections flow seamlessly.

Reviewer comment: In the discussion and conclusion the authors should investigate the difference between support staff’s opinions and the opinion of service managers’ and trainers’.

Our response: There was no major discrepancy in the views of support staff and the service managers/PBS trainers. We have now clarified this in the Conclusion section. 

Round 2

Reviewer 1 Report

Thank you for addressing the suggested revisions. With these changes in place I believe the paper is suitable for publication.

Author Response

Thanks. We have made the changes.

Reviewer 2 Report

Reviewer comment: 2. Please complete the Introduction with references to E. Emerson, who has contributed significantly to understanding the issue in question.

Our response: We have now provided Emerson’s reference to define Behaviours that challenge.

One work by Emerson from 1995 was used. Please refer to more recent work. Emerson does not use the term “Behavior that challenge (BtC)”, so it is not justified to refer to his work here (Introduction, p. 1).

Reviewer comment: 5. The authors focus on two topics that resulted from the thematic analysis. This approach is rare in qualitative research. All the themes that emerged during data analysis constitute a whole that describes the essence of the phenomenon. I believe that a lot of space in the work is devoted to the description of the program - also in the description of the results. It is impossible to present research results and the program simultaneously. I propose to present all the results and mention the program in the Conclusions, devoting a separate article to it. Interjected program descriptions create cognitive chaos.

Our response: We agree with the reviewer. However, the emerging themes although interconnected but could be categorised under two distinct subthemes, one including ‘the causes of behaviour’ and ‘alternatives to medication to manage behaviour’, and the other one including ‘the current trend in medication use’ and ‘withdrawal of medication.’ The first subtheme covers the “behaviours,” and the second subtheme covers the “medication.” Each of them needs a separate Introduction and

Discussion. Each of them has a wealth of information including many subthemes which would be difficult to fit within one paper given the restricted space. Also given the distinct nature of these two themes presenting them together would have confused the readers and as a result, very important messages concerning these two distinct themes would have been lost.

Our response: We have now moved SPECTROM discussions from the Results section to the Discussion section.

SPECTROM descriptions are still in the Results section.

Please clearly indicate the main themes and subthemes in the Results section so that there is no doubt about it. Currently, the text shows that there may be 6 main themes, not 2.

Author Response

We have clarified that there were two main themes and six subthemes. To reflect this, we have converted the subthemes to Italics from bold to distinguish them from the main themes that are presented in bold.